# Numerical Simulation on Aerodynamic Characteristics of Transition Section of Tilt-Wing Aircraft

**Qingjin Huang** **, Guoyi He \*, Jike Jia, Zhile Hong and Feng Yu**

School of Aircraft Engineering, Nanchang Hangkong University, Fenghe Nan Avenue 696, Nanchang 330063, China; 2106085800019@stu.nchu.edu.cn (Q.H.); 2106085500007@stu.nchu.edu.cn (J.J.); 2206085500005@stu.nchu.edu.cn (Z.H.); 2106085500021@stu.nchu.edu.cn (F.Y.)

\* Correspondence: 70190@nchu.edu.cn

**Abstract:** The tilt-wing aircraft has attracted widespread attention due to its excellent performance. However, its aerodynamic characteristics during the tilt transition section are characterized by unsteadiness, nonlinearity, and strong coupling, making it difficult to control. Using computational fluid dynamics (CFD) methods and moving overset grids to control the tilt-wing motion, the momentum source method is employed to replace actual propellers. The influence of the propeller on the aerodynamic characteristics of the tiltrotor at different tilt angles is investigated under incoming flow velocities of 8 m/s and 45 m/s in steady conditions. Additionally, the differences between steady and unsteady calculations of the tilt transition section are investigated at incoming flow velocities of 8 m/s, 15 m/s, 30 m/s, and 45 m/s in unsteady conditions. The research results indicate the following information: 1. the slipstream from the propellers significantly enhances the lift, drag, and stall angle of attack of the tilt-wing aircraft but reduces the lift-to-drag ratio; 2. there are noticeable differences in the forces acting on the tilt-wing aircraft between steady calculations with fixed tilt angles and unsteady calculations with continuous tilting.

**Keywords:** tilt-wing aircraft; CFD; tilt transition section; momentum source method

## 1. Introduction

With the advancement of urbanization, ground transportation is facing increasing pressure, and traffic congestion has become a major issue in many large cities. Urban Air Mobility (UAM) has received a lot of attention in a number of programs [1]. With the electrification of the aviation industry [2], electric vertical take-off and landing vehicles (eVTOLs) have become a major option for UAM [3–5]. Currently, eVTOLs have various aerodynamic configurations, including multi-rotor configuration (Figure 1a), compound propulsion configuration (Figure 1b), tiltrotor configuration (Figure 1c), etc.

Among these configurations, the multi-rotor and compound propulsion configurations have simple structures, are easy to control, easy to manufacture, and cost-effective, making them the preferred choices for many eVTOL companies. However, the multi-rotor and compound propulsion configurations have low aerodynamic efficiency and are constrained by the current low energy density of battery technology, which inevitably leads to losses in the flight speed, range, and payload of eVTOL aircraft. Therefore, the multi-rotor configuration is not suitable for long-distance flights. The compound propulsion configuration slightly improves aerodynamic efficiency compared to the multi-rotor configuration, but its performance limit still cannot surpass the tiltrotor configuration [6].

The tiltrotor configuration vehicle is capable of vertical take-off and landing and also has the advantages of fast speed and a large load capacity of fixed-wing airplanes [7–9], so it has received wide attention among many eVTOL aerodynamic configurations. However, during the take-off and landing section of a tiltrotor aircraft, the high-speed slipstream of the propeller will impact the wing, resulting in a loss of available power [10,11], so the tilt-wing

configuration was developed based on the tiltrotor. Although a tiltrotor aircraft has obvious advantages, the aerodynamic characteristics in the tilt transition section are extremely complex, posing a challenge to the control of the aircraft [12–14]. To achieve smooth control of the tilt-wing aircraft in the tilt transition section, it is particularly important to study the aerodynamic characteristics of the aircraft in this stage.

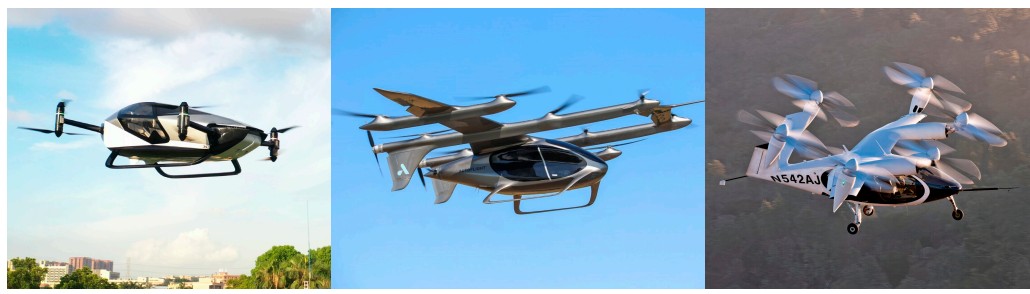

(**a**) Multi-rotor configuration    (**b**) Compound propulsion configuration    (**c**) Tiltrotor configuration

**Figure 1.** Three different configurations of eVTOL.

Research on the aerodynamic characteristics of tiltrotor aircraft is mainly conducted using experimental methods [15–17] and numerical simulations [18,19]. However, due to the long experimental research cycle and high costs associated with experiments, with the rapid development of computer technology, CFD simulation has become an important method for studying the aerodynamic characteristics of tiltrotor aircraft [20,21]. In earlier years, numerous European institutions developed various high-fidelity CFD algorithms for simulating and analyzing the aerodynamic characteristics of tilt-wing aircraft [22–24]. These studies demonstrated the advantages of tilt-wing aircraft over tiltrotor aircraft. In recent years, scholars have also conducted research on the aerodynamic characteristics of the XV-15 using high-fidelity CFD [25–28]. However, the research on tiltrotor aircraft using high-fidelity CFD methods is limited by computer performance and computational costs, and most studies only focus on a specific state of the tiltrotor, such as the helicopter mode or fixed-wing mode. However, research on continuous tilt transition is relatively scarce, and most studies decompose the continuous tilt transition process into multiple fixed tilt angles, using quasi-steady calculations [29,30], which cannot effectively simulate the unsteady aerodynamic characteristics of the continuous tilt transition process. The limited simulation of unsteady conditions is mainly attributed to the intricate aerodynamic calculations required for propellers and the substantial computational demands when considering the combined effects of tilt, propeller aerodynamics, and tilt-wing interactions. In tilt-wing aircraft, the wings tilt along with the propeller, resulting in extremely complex stall problems for the tilt-wings during the tilt transition section, which leads to even more complex aerodynamic characteristics during the tilt transition section. If high-fidelity CFD methods are used to simulate this process, the computational time and cost may be unacceptable.

In this paper, the momentum source method is used to replace the blade-resolved CFD, simplifying the calculations. The URANS method is used to simulate the aerodynamic characteristics of the tilt-wing aircraft with continuous tilting of the propeller/wing assembly and compare it with the quasi-steady state calculated by the RANS method. The aerodynamic efficiency of continuous tilting conditions is obtained, providing references for the design of tilt-wing aircraft and the acquisition of aerodynamic data at this stage.

## 2. Methodological Description

### 2.1. Computational Model

The computational model consists of a certain model eVTOL main wing/rotor combination, as shown in Figure 2a,b. Figure 2a depicts the combination of the propeller and

wing, with a half-wing span of 6.204 m, fixed wing length of 3.279 m, tilt-wing length of 2.925 m, NACA4415 airfoil for the wing, root chord length of 1.4 m, tip chord length of 0.85 m, and a wing installation angle of 3°. The propeller radius is 1.45 m, with VR5 blade airfoil. The twist and chord distribution of the propeller blades are shown in Table 1. In the table, 'r' represents the distance of the section from the propeller center, and 'R' is the propeller radius. Figure 2b illustrates the actual model used for computation, where the propeller is replaced by the momentum source method, requiring grid refinement only at the corresponding propeller positions. The grid refinement region is a cylinder with a radius of 1.5 m and a height of 0.12 m.

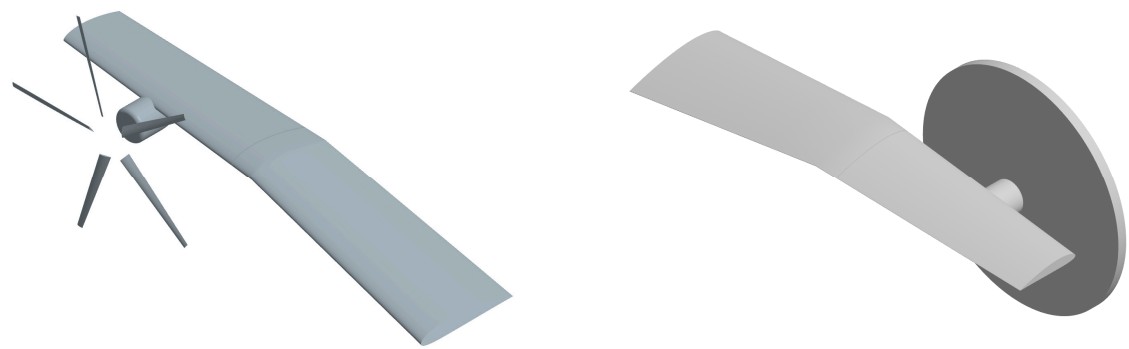

(**a**) Unsimplified propeller/wing combination.   (**b**) Simplified propeller/wing combination.

**Figure 2.** Illustration of the computational model.

**Table 1.** Modeling data for propeller blades.

| Positioning (r/R) | Twist (deg) | Chord Length (m) |
|---|---|---|
| 0.1862 | 35.000 | 0.141 |
| 0.25 | 33.001 | 0.136 |
| 0.35 | 31.434 | 0.129 |
| 0.45 | 26.734 | 0.121 |
| 0.55 | 23.601 | 0.114 |
| 0.65 | 20.467 | 0.106 |
| 0.75 | 17.333 | 0.099 |
| 0.85 | 14.200 | 0.091 |
| 0.95 | 11.067 | 0.083 |
| 1 | 9.500 | 0.080 |

The schematic diagram of the computational domain and boundary conditions is shown in Figure 3. The size of the computational domain is 70 m × 42 m × 80 m, with the fixed section of the wing maintaining a constant angle of attack of 0°. The tilt section undergoes tilting, with a tilt angle of 0° defined in the fixed-wing aircraft mode and 90° in the helicopter mode. The incoming flow velocity remains constant during the tilting process and is set to 8 m/s, 15 m/s, 30 m/s, and 45 m/s. The inlet type is velocity inlet, the outlet type is pressure outlet, the wing root plane is set as symmetry, and the remaining surfaces are slipwalls.

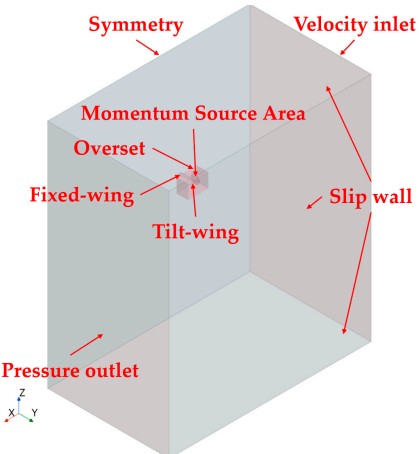

**Figure 3.** Illustration of the computational domain and boundary condition division.

*2.2. Numerical Simulation Methods*

In this paper, the computational software used is STAR-CCM+ 2302.0001, and its CFD method is based on the finite volume method. The finite volume method discretizes the solution domain into a finite number of small control volumes using computational grids. When performing finite volume integration, the governing equations for fluid, including the continuity equation, momentum equation, and energy equation, can be written as follows:

$$\frac{\partial}{\partial t}\int_V \rho \cdot dV + \oint_A \rho v \cdot da = \int_V S_u \cdot dV \tag{1}$$

In Equation (1), $t$ represents time; $V$ represents volume; $a$ represents the area vector; $\rho$ represents density; $v$ represents velocity; $S_u$ represents the source term.

$$\frac{\partial}{\partial t}\int_V \rho v \cdot dV + \oint_A \rho v \otimes v \cdot da = -\oint_A pI \cdot da + \oint_A T \cdot da + \int_V f_b \cdot dV + \int_V S_u \cdot dV \tag{2}$$

In Equation (2), $p$ represents pressure; $I$ represents the normal stress tensor; $T$ represents the viscous stress tensor; $f_b$ represents the sum of volume forces; $S_u$ represents the source term.

$$\frac{\partial}{\partial t}\int_V \rho E \cdot dV + \oint_A \rho H v \cdot da = -\oint_A q \cdot da + \oint_A Tv \cdot da + \int_V f_b v \cdot dV + \int_V S_u \cdot dV \tag{3}$$

In Equation (3), $E$ represents the total energy; $H$ represents the total enthalpy; $q$ represents the heat flux.

The addition of appropriate constitutive relationships can make the system of equations closed. For these control volumes, integrating the general transport equations over control volume $V$ yields the integral form of the transport equations.

$$\frac{d}{dt}\underbrace{\int_V \rho\phi dV}_{Transient\ Term} + \underbrace{\int_A \rho v\phi da}_{Convective\ Flux} = \underbrace{\int_A \Gamma\nabla\phi da}_{Diffusive\ Flux} + \underbrace{\int_V S_\phi dV}_{Source\ Term} \tag{4}$$

Considering the convergence during computation, this paper adopts a coupled solver. The coupled solver solves the continuity equation, momentum equation, and energy equation simultaneously. The velocity of the flow field is obtained from the momentum equation, the pressure is obtained based on the continuity equation, and the density is obtained through the equation of state. For any control volume with a differential surface

area *da* and volume *V*, the vector form of the fluid control equations in the Cartesian coordinate system can be written as follows:

$$\frac{\partial}{\partial t}\int_V W \cdot dV + \oint [F - G] \cdot da = \int_V H \cdot dV \tag{5}$$

In Equation (5),

$$W = \begin{bmatrix} \rho \\ \rho v \\ \rho E \end{bmatrix} \tag{6}$$

$$F = \begin{bmatrix} \rho v \\ \rho v^2 + pI \\ \rho vH + pv \end{bmatrix} \tag{7}$$

$$G = \begin{bmatrix} 0 \\ T \\ Tv + \dot{q} \end{bmatrix} \tag{8}$$

$$H = \begin{bmatrix} S_u \\ f_r + f_g + f_p + f_u + f_w + f_L \\ S_u \end{bmatrix} \tag{9}$$

In Equations (6)–(9), $\rho$ represents fluid density; $v$ represents fluid velocity; $p$ represents fluid pressure; $E$ represents the total energy per unit mass of the fluid; $T$ represents the viscous stress tensor; $\dot{q}$ represents the heat flux vector; $H$ represents the volume force vector. The relationship between $E$ and $H$ can be expressed as follows:

$$E = H - \frac{p}{\rho} \tag{10}$$

$$H = h - \frac{|v|^2}{2} \tag{11}$$

$$h = C_p T \tag{12}$$

In Equation (12), $C_p$ represents the specific heat capacity, and $T$ represents temperature.

The research content of this paper is at low Mach numbers, where the fluid can be treated as incompressible, thus introducing a preconditioning equation:

$$\Gamma \frac{\partial}{\partial t}\int_V Q \cdot dV + \oint [F - G] \cdot da = \int_V H \cdot dV \tag{13}$$

In Equation (13),

$$\Gamma = \begin{bmatrix} \theta & 0 & \rho_T \\ \theta v & pI & \rho_T \\ \theta H - \delta & pv & \rho_T H + \rho C_p \end{bmatrix} \tag{14}$$

In Equation (14), $\rho_T$ represents the derivative of density with respect to temperature at a constant pressure:

$$\rho_T = \left.\frac{\partial \rho}{\partial T}\right|_p \tag{15}$$

In this study, the fluid chosen is an ideal gas; therefore, the following equations can be obtained:

$$\delta = 1 \tag{16}$$

$$\rho_T = -\frac{p}{RT} \tag{17}$$

$$\theta = \frac{1}{U_r^2} - \frac{\rho_T}{\rho C_p} \tag{18}$$

In Equation (18),

$$U_r = \min\left[\max\left(|v|, \frac{\nu}{\Delta x}, \varepsilon\sqrt{\frac{\delta p}{\rho}}, U_{r\ min}\right), U_{r\ max}\right] \tag{19}$$

In Equation (19), $\Delta x$ represents the length scale between grid cells where diffusion occurs; $\delta p$ is the pressure difference between adjacent grid cells; $\varepsilon = 2$.

By applying Equation (13) to the grid cell O, the following system of discrete equations can be obtained:

$$V_O \Gamma_O \frac{\partial Q_O}{\partial t} + \sum_f \left(f_f - g_f\right) \cdot a = h V_O \tag{20}$$

In Equation (20), $f_f$ represents the inviscid flux through face f; $g_f$ represents the viscous flux through face f; $V_O$ denotes the volume of grid cell O; $\Gamma_O$ represents the preconditioning matrix of grid cell O.

For the inviscid flux, using the Weiss–Smith preprocessed Roe flux splitting scheme, $f_f$ can be expressed as follows:

$$f_f = \frac{1}{2}(f_0 + f_1) - \frac{1}{2}\Gamma |A| \Delta Q \tag{21}$$

In Equation (21), subscripts 0 and 1 represent arbitrary grid cells on either side of the $f$-face; $\Gamma$ is the preconditioning matrix.

$$\Delta Q = (Q_1^r - Q_0^r) \tag{22}$$

$$|A| = M|\Lambda|M^{-1} \tag{23}$$

In Equation (23), $\Lambda$ is a diagonal matrix of eigenvalues, and $M$ is the modal matrix of diagonalized $\Gamma^{-1}(\partial f/\partial Q)$.

For steady calculations, Equation (5) can be discretized in time as follows:

$$\left[D_i + \sum_j^{N_{faces}} S_{j,k}\right] \Delta Q = -R_i^n \tag{24}$$

In Equation (24),

$$D_i = \frac{V}{\Delta t}\Gamma + \sum_j^{N_{faces}} S_{j,k} \tag{25}$$

$$S_{j,k} = \frac{\partial F_j}{\partial Q_k} - \frac{\partial G_j}{\partial Q_k} \tag{26}$$

$$\Delta Q \equiv Q^{n+1} - Q^n \tag{27}$$

$$R_i^n = \sum_j^{N_{faces}} \left(F_j - G_j\right)^n \tag{28}$$

For unsteady calculations, this paper investigates the use of an implicit time-stepping method. Since the time derivative preconditioning compromises the temporal accuracy of the governing equations, a pseudo-time derivative term is introduced in Equation (5):

$$\frac{\partial}{\partial t} \int_V W \cdot dV + \Gamma \frac{\partial}{\partial \tau} \int_V Q \cdot dV + \oint [F - G] \cdot da = \int_V H \cdot dV \qquad (29)$$

In Equation (29), $t$ represents the physical time step, and $\tau$ represents the pseudo time step. As $\tau$ iterates, the pseudo time derivative term in Equation (29) leans towards 0, and Equation (29) reduces to Equation (5).

The interaction of the propeller with the air is replaced by the time-averaged momentum source term distributed across the swept region of the propeller. A small element on the rotor disk with an area of $S_\Delta$, located at a distance r from the center of the rotor disk, with a chord length of $dr$, and corresponding azimuthal angle of $\Delta\psi$, is selected as the research object. If the force experienced by this small element is $dF$, then the force exerted by it on the air is $-dF$. Assuming the number of rotor blades is $N$, the total force exerted on the air during one revolution is $-N \cdot dF$. The time-averaged force exerted on the element can be expressed as follows:

$$F_\Delta = \frac{N \cdot (-dF)}{2\pi r dr} S_\Delta \qquad (30)$$

$F_\Delta$ represents the momentum source term, which can be decomposed and substituted into the momentum source term of the flow field control equation to simulate the effect of the rotor on the air.

In this study, we have selected the $k$-$\Omega$ SST turbulence model [31]. This specific model is widely acknowledged and extensively utilized for various turbulence calculations. It has demonstrated high accuracy in predicting flow behavior, even in scenarios with significant adverse pressure gradients and high rotational speeds.

### 2.3. Numerical Method Validation

Numerical methods for calculating propeller aerodynamics include the momentum source method [32,33], the multiple reference frame model [34,35], the sliding mesh [36,37], and the moving overset grids [38,39]. These methods exhibit increasing computational complexity. In this paper, our focus is solely on investigating the impact of propeller slipstream on the wing during the transition section, without delving into the intricate aerodynamic characteristics of the propeller itself. Consequently, to minimize computational complexity, we have opted for the momentum source method to simulate the propeller's influence on the airflow.

To validate the feasibility of the momentum source method, a comparison was made using the isolated rotor test data from reference [40]. The detailed parameters of the rotor in reference [40] are presented in Table 2. Experimental measurements of dynamic pressure were conducted along two straight lines beneath the rotor, namely at 0.215 R and 0.660 R (where R represents the radius of the rotor as used in reference [40]), with a length of 1.3 R.

**Table 2.** Parameters for isolated rotor validation test cases.

| Parameter | Data |
| --- | --- |
| Rotor airfoil | NACA0012 |
| Number of rotor blades | 2 |
| Rotor blade installation angle (deg) | 11 |
| Rotor radius (m) | 0.914 |
| Rotor blade chord length (m) | 0.1 |
| Rotor rotation speed (rad·s$^{-1}$) | 122.2 |
| Rotor height above ground | 3.6 R |

Figure 4a,b show the dynamic pressure comparisons at 0.215 R and 0.660 R below the rotor. The momentum source method accurately captures the trend of the rotor slipstream, consistent with the experimental results. However, the presence of the testing apparatus at 0.660 R caused flow blockage, which disrupted the test data.

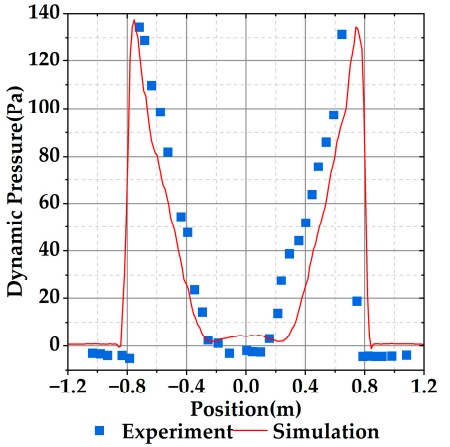 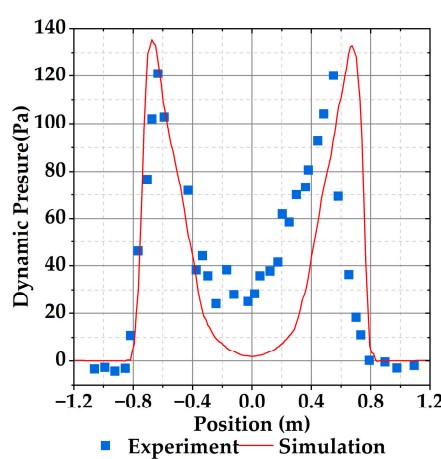

(**a**) Variation curve of dynamic pressure with position.

(**b**) Variation curve of dynamic pressure with position.

**Figure 4.** Comparison of dynamic pressure below the rotor at 0.215 R and 0.660 R.

To further validate the reliability of the momentum source used in this paper, calculations were performed on the propeller/wing combination studied in this paper using a multiple reference frame model, sliding mesh, and momentum source method. The calculated data include propeller thrust, wing lift, and wing drag. The comparison was made under the following calculation conditions: a tilt angle for the tilt-wing of 0°, angle of attack for the fixed wing of 0°, incoming flow velocity of 8 m/s and propeller rotation speed of 1400 RPM. The calculated results are shown in Table 3.

**Table 3.** Propeller thrust, wing lift, and drag calculated using three methods.

| Calculation Method | Propeller Thrust/(N) | Lift/(N) | Drag/(N) |
| --- | --- | --- | --- |
| Multiple reference frame | 5990.522 | 575.292 | 41.140 |
| Sliding grid | 5988.837 | 584.166 | 44.316 |
| Momentum source method | 6531.764 | 591.618 | 46.331 |

As seen from the data in Table 3, the wing lift and drag computed by the momentum source method only slightly deviate from that calculated by the multiple reference frame model and the sliding mesh. Therefore, it can be concluded that the momentum source method remains effective in accounting for the impact of the propeller slipstream on the wing.

To ensure that the computational grid does not affect the results, grid-independence validation data are given in Table 4. The boundary conditions of the validation condition are as follows: an incoming flow velocity of 8 m/s; propeller speed of 1400 RPM; wing angle of attack of 0°; tilt angle of 0°. Combining the data tolerance situation and calculation time, the grid with 13,504,231 calculation grid cells is selected as the calculation grid.

As shown in Figure 5a,b, the time-step independence verification was conducted under the return transition section condition, with an incoming flow velocity of 8 m/s, propeller rotation speed of 1400 RPM, tilt completion time of 10 s, and a total of 13,504,231 grid cells used for the calculations. The lift and drag of the wing were calculated at three different time steps. The compared time steps were 0.001 s, 0.0045 s, and 0.009 s. Based on computational speed and convergence, the final chosen time step for calculation was 0.0045 s.

**Table 4.** Grid-independent validation data.

| Grid Cell Number | Lift | Drag |
|---|---|---|
| 4,809,309 | 615.58 | 50.51 |
| 7,944,514 | 611.10 | 49.97 |
| 9,074,088 | 600.70 | 47.21 |
| 11,748,786 | 591.10 | 45.97 |
| 13,504,231 | 591.62 | 46.33 |
| 19,065,387 | 592.52 | 46.21 |

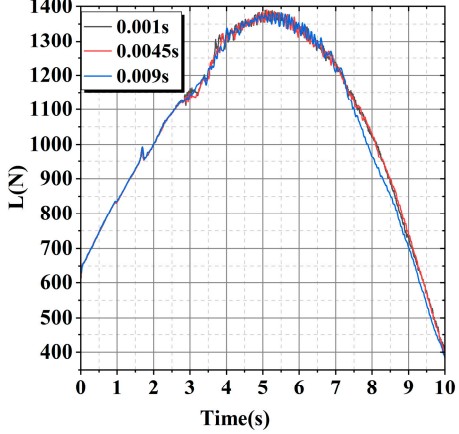

(**a**) Variation curve of lift with time.

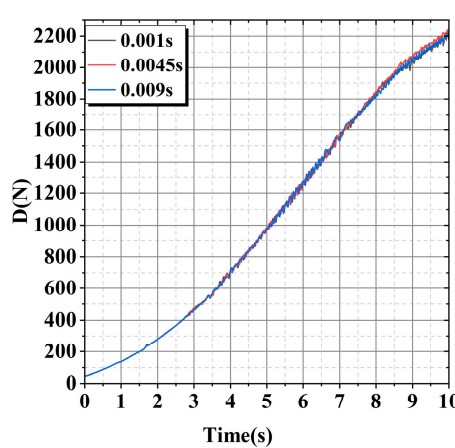

(**b**) Variation curve of drag with time.

**Figure 5.** Comparison of time-step independence verification.

## 3. Results and Analyses

In this study, both steady calculations with a fixed tilt angle and unsteady calculations with continuous tilting were used to investigate the aerodynamic characteristics of the wing during the transition section.

Using steady calculations, we investigate the influence of a propeller on the aerodynamic characteristics of a tilted wing during the transition section. The wing is defined to have a tilt angle of 0° in the fixed-wing mode and 90° in the helicopter mode. The tilt angle varies from 0° to 90° in increments of 10° for the calculations. When the propeller is present, it operates at a constant speed of 1400 RPM with a fixed pitch. The freestream velocities considered are 8 m/s and 45 m/s.

Using unsteady calculations, we investigate the aerodynamic characteristics of a tilted wing during continuous transition of a tiltrotor aircraft. Due to the actual tilting of the wing, the results of unsteady calculations are divided into the take-off transition section and return transition section. The time taken to complete the transition is 5 s and 10 s, respectively, with the wing tilting at a constant speed. The propeller operates at a speed of 1400 RPM with a fixed pitch. The freestream velocities considered are 8 m/s, 15 m/s, 30 m/s, and 45 m/s.

### 3.1. Impact of Propeller on Tilt-Wing

Figures 6 and 7 represent the variation in the lift coefficient and drag coefficient of the wing with an angle of attack at flow velocities of 8 m/s and 45 m/s, respectively. Figures 8 and 9 show the lift-to-drag ratio of the wing at flow velocities of 8 m/s and 45 m/s, respectively. From the figures, it can be observed that the addition of the propeller significantly increases the lift coefficient, drag coefficient, and stall angle of the wing. However, the lift-to-drag ratio decreases, as shown in Figures 8 and 9.

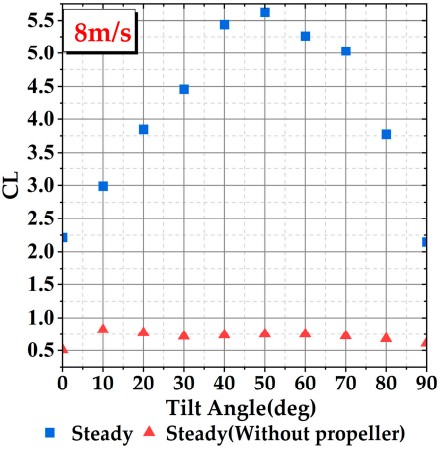

(**a**) Lift coefficient at different tilt angles.

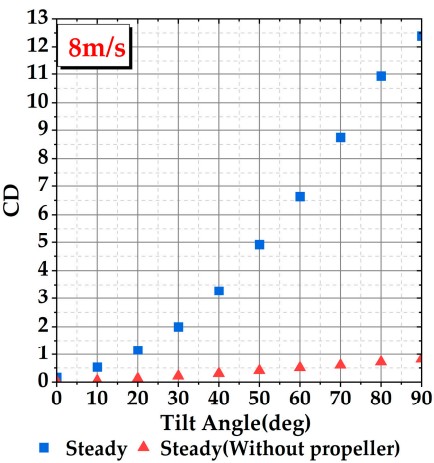

(**b**) Drag coefficient at different tilt angles.

**Figure 6.** Comparison of lift/drag coefficients at each tilt angle of the wing with and without a propeller for an incoming flow velocity of 8 m/s.

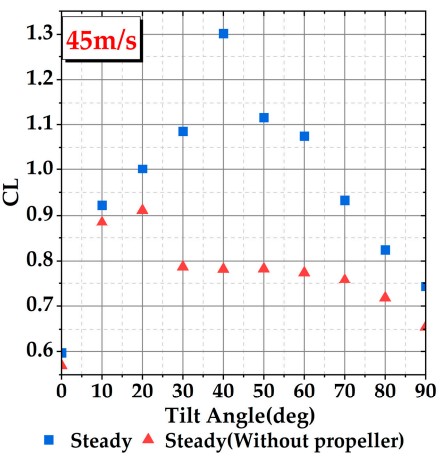

(**a**) Lift coefficient at different tilt angles.

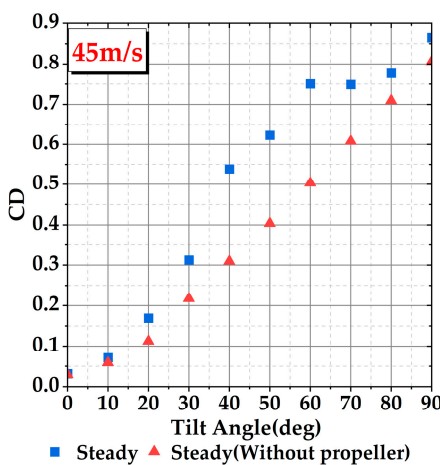

(**b**) Drag coefficient at different tilt angles.

**Figure 7.** Comparison of lift/drag coefficients at each tilt angle of the wing with and without a propeller for an incoming flow velocity of 45 m/s.

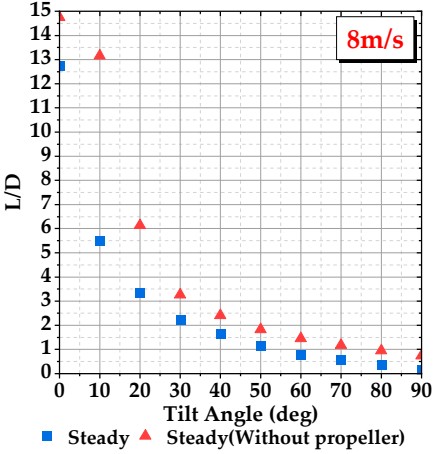

**Figure 8.** Comparison of lift-to-drag ratio between the wing with propeller and wing without propeller at various tilt angles with an incoming flow velocity of 8 m/s.

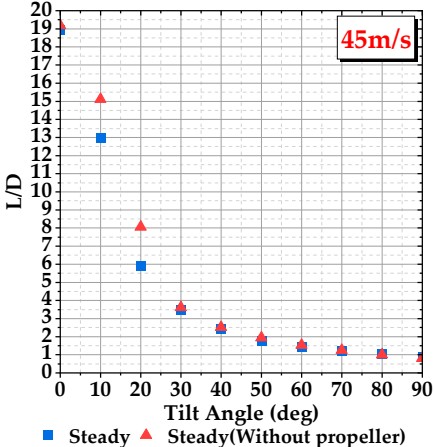

**Figure 9.** Comparison of lift-to-drag ratio between wing with propeller and wing without propeller at various tilt angles with an in-coming flow velocity of 45 m/s.

The increase in lift coefficient, drag coefficient, and stall angle is due to the slipstream effect of the propeller. For example, an incoming flow speed of 8 m/s and a tilt angle of 40° can be used. In the research process of this paper, when the aircraft is in the fixed-wing mode, the angle of attack is always 0°. After the tilt-wing starts to tilt, the tilt angle of the tilt-wing is equal to the angle of attack of the tilt-wing. A section of the tilting wing can be taken for analysis, as shown in Figure 10, which illustrates the flow and force on this section. Analysis of the figure reveals that due to the tilting wing tilting together with the propeller, the slipstream $V_R$ of the propeller combined with the incoming flow $V$ results in a deflection of the actual flow direction $V_T$ of the tilting wing. According to the definition of aerodynamics, the lift and drag of this section can be represented as $L_T$ and $D_T$ in Figure 10. However, in actual research, considering the aircraft as a whole, the actual directions of lift and drag should be represented as $L$ and $D$ in Figure 10. Decomposing the lift and drag in the local airflow coordinate system into the actual lift and drag directions reveals that a portion of the lift $L_T$ and drag $D_T$ in the local airflow coordinate system is decomposed into the drag of the aircraft airflow coordinate system, thereby causing the rate of drag increase to be faster compared to the rate of drag increase when the angle of attack changes in a typical fixed-wing aircraft.

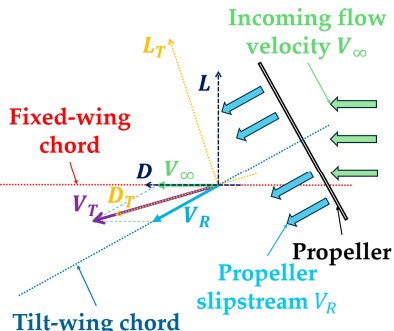

**Figure 10.** Schematic diagram of aerodynamic force offset in the tilting wing section.

Additionally, due to the propeller slipstream causing a deflection in the actual flow direction of the tilting wing, the actual angle of attack of the tilting wing section is smaller than the tilt angle. As a result, the overall aerodynamic characteristics of the aircraft manifest in a significant increase in the wing stall angle of attack. As shown in Figure 11a,b, without the propeller, the tilting wing section has already stalled at that tilt angle, and the flow on the upper surface of the tilting wing is in a turbulent state; however, with the propeller slipstream, the airflow attachment on the upper surface of the tilting wing remains relatively stable.

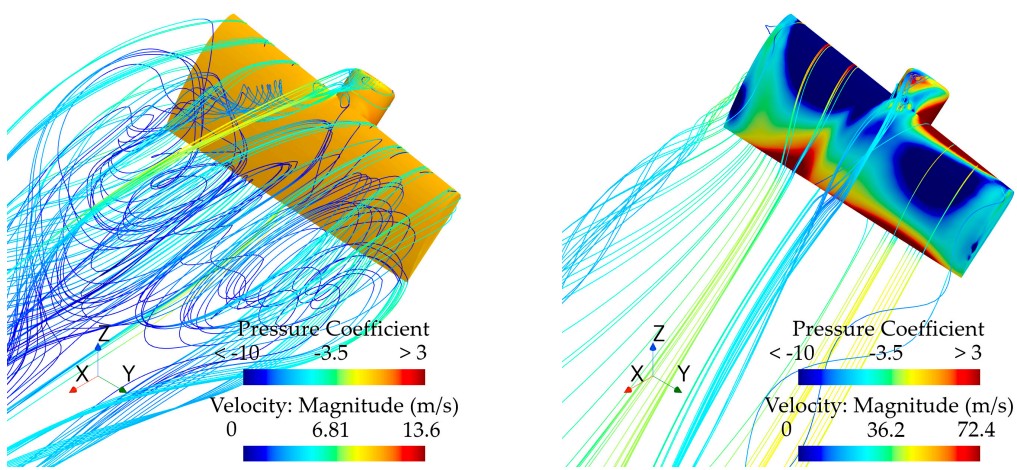

(**a**) Surface pressure coefficient contour and wing streamline.

(**b**) Surface pressure coefficient contour and wing streamline.

**Figure 11.** Comparison of surface pressure coefficient contour and wing streamline on the wing with and without propeller at a tilt angle of 40°.

As shown in Figure 11, the slipstream of the propeller generates additional lift and drag. However, since it also changes the actual angle of attack of the wing, the additional lift and drag are shifted rearward. After decomposing the additional lift and drag into the actual lift and drag directions, the components of the additional aerodynamic forces result in a significant increase in the overall drag of the wing, thereby reducing the overall lift-to-drag ratio.

*3.2. Aerodynamic Characteristics of the Tilt Transition Section*

3.2.1. Explanation of Calculation Results

The steady calculation with a fixed tilt angle does not require distinguishing between the take-off transition section (TTS) and return transition section (RTS). Furthermore, as the tilt angle remains constant, the forces at each tilt angle are not influenced by time. The continuous unsteady calculations for tilting are conducted using two distinct sets of curves for the take-off and return transitions to realistically simulate the process of take-off and landing transitions. Due to the actual tilting process, the wing force curves vary for different tilting durations. To compare the steady calculations with the unsteady calculations, the data obtained from the unsteady state calculations were transformed from lift/drag versus time curves to lift/drag versus tilt angle curves. Figures 12–15 show the wing lift/drag coefficient versus the tilt angle for incoming flow velocities of 8 m/s, 15 m/s, 30 m/s, and 45 m/s, respectively.

In this study, the take-off and return transition sections are designed to be fully reversible in terms of the motion process. However, when comparing the forces on the wing during the two processes, a significant difference is observed in the transition section between take-off and return. Comparing the results of steady calculations with unsteady calculations, it is found that the maximum values of lift and drag in the return transition section are higher than those obtained from steady calculations. The values of wing lift and drag in the take-off transition section at 8 m/s and 15 m/s, as well as their trends, are similar to those obtained from steady calculations. When the incoming flow velocity is 30 m/s and the tilting time is 5 s, the trends of lift and drag changes of the wing are closer to the steady-state calculation results. However, when the tilting time is 10 s, there is a significant difference in lift and drag compared to 5 s in the tilting angle range of 40° to 70°. This phenomenon is caused by the leading-edge vortex excited by the leading edge of the tilting wing, which will be explained in more detail in the following sections. At an incoming flow velocity of 45 m/s, the maximum lift in the take-off transition section is significantly lower than the maximum lift obtained from steady calculations. Conversely,

the maximum values of lift and drag in the return transition section are higher than those calculated in the steady state.

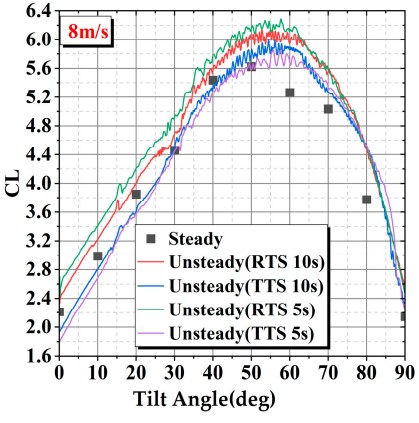

(**a**) Lift coefficient at different tilt angles.

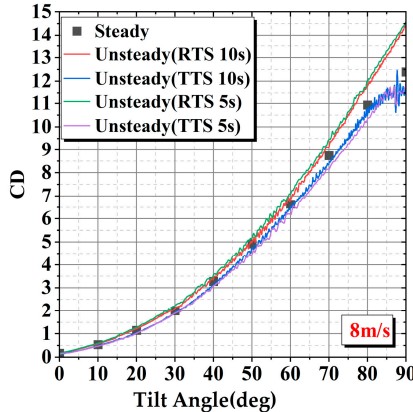

(**b**) Drag coefficient at different tilt angles.

**Figure 12.** Comparison of lift/drag coefficient variation with tilt angle between steady and unsteady calculations at an incoming flow velocity of 8 m/s.

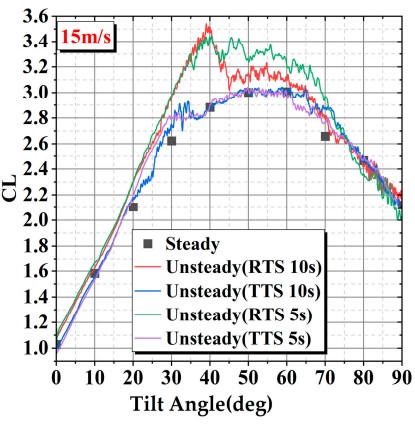

(**a**) Lift coefficient at different tilt angles.

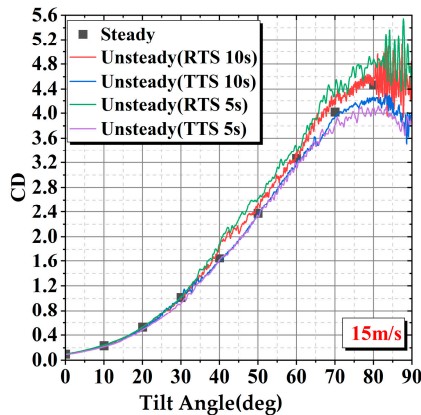

(**b**) Drag coefficient at different tilt angles.

**Figure 13.** Comparison of lift/drag coefficient variation with tilt angle between steady and unsteady calculations at an incoming flow velocity of 15 m/s.

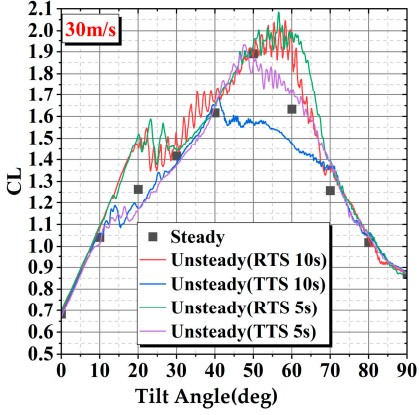

(**a**) Lift coefficient at different tilt angles.

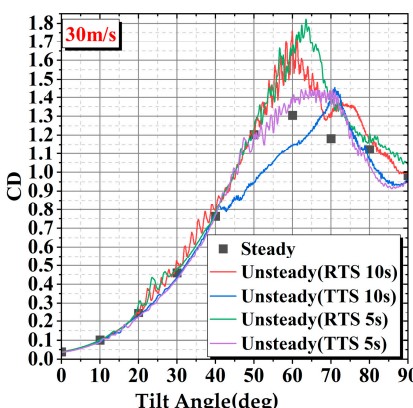

(**b**) Drag coefficient at different tilt angles.

**Figure 14.** Comparison of lift/drag coefficient variation with tilt angle between steady and unsteady calculations at an incoming flow velocity of 30 m/s.

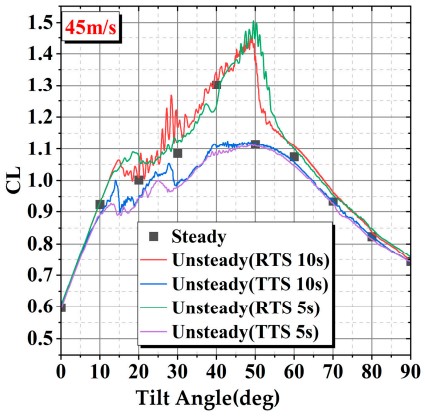
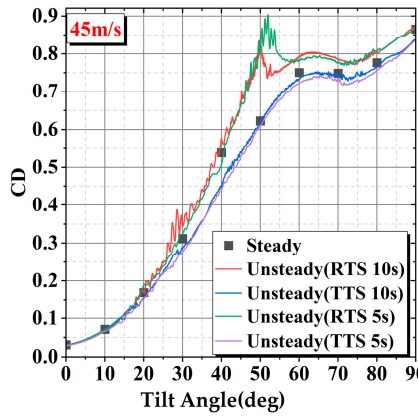

(**a**) Lift coefficient at different tilt angles.  (**b**) Drag coefficient at different tilt angles.

**Figure 15.** Comparison of lift/drag coefficient variation with tilt angle between steady and unsteady calculations at an incoming flow velocity of 45 m/s.

3.2.2. Differences in Aerodynamic Forces between Take-Off Transition and Return Transition Section

The pressure on the wing surface can intuitively reflect the magnitude of the aerodynamic force on the wing. Therefore, for the issue of differences in aerodynamic forces during the take-off and return transition sections of the tilt-wing aircraft, the wing surface pressure coefficient contour can be used for preliminary assessment. As shown in Figure 16a,b, the pressure coefficient contour on the upper surface of the wing during the take-off and return transition sections is presented at a freestream velocity of 15 m/s and a tilt angle of 40.5°. From the graphs, it can be observed that during the return transition section, the tilt-wing has stronger negative pressure at the left end of the power compartment. Consequently, it generates a greater aerodynamic force.

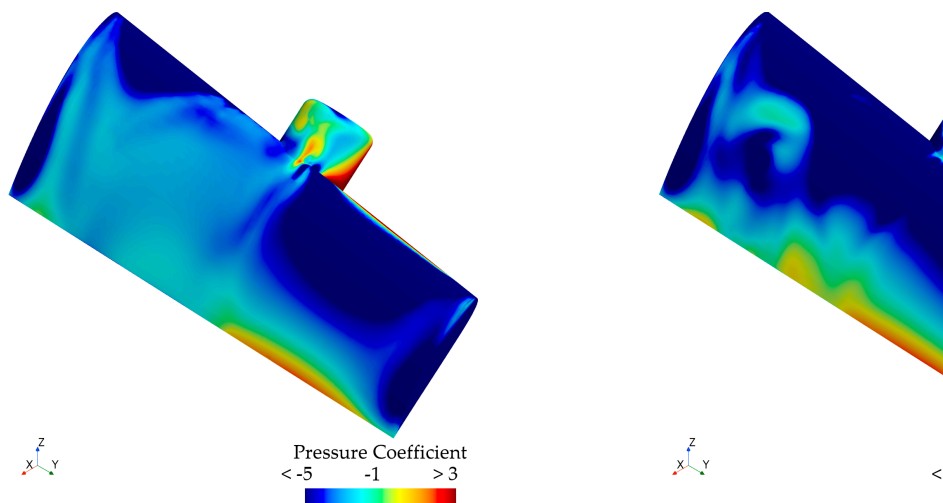

(**a**) Pressure coefficient on the upper surface of the tilt-wing during take-off transition section.  (**b**) Pressure coefficient on the upper surface of the tilt-wing during return transition section.

**Figure 16.** Pressure coefficient contour on the upper surface of the wing during the take-off and return transition section, with an incoming flow velocity of 15 m/s and a tilt angle of 40.5°.

To investigate the cause of the phenomenon shown in the above figure, the Q criterion is used to plot the vorticity distribution on the wing, and the vorticity surfaces are rendered using velocity. As shown in Figure 17a,b, the vorticity distributions on the wing during the

take-off and return transition sections at a tilt angle of 40.5° are presented. By comparing the two figures, it can be observed that in Figure 17a, at the location where the wing's negative pressure is lower, the vortices appear to break and detach. This indicates severe airflow separation on the wing surface at that location, leading to a decrease in flow velocity and a weakening of the negative pressure intensity in that area. On the other hand, in Figure 12b, separation vortices are also present on the left side of the wing, but they do not detach significantly from the wing surface. Therefore, the flow velocity on the wing surface is maintained at that location, resulting in a higher negative pressure intensity compared to the same position in Figure 12a.

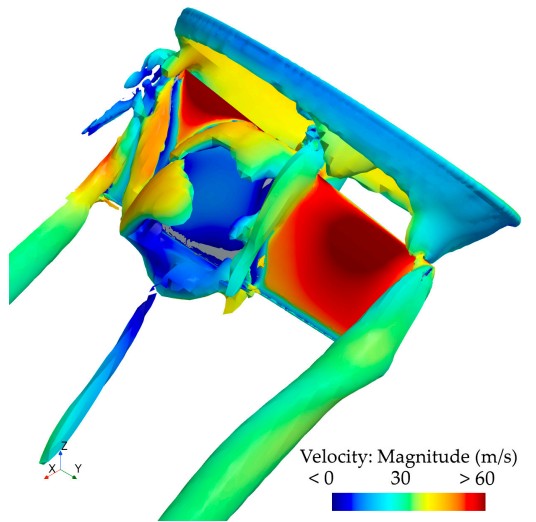
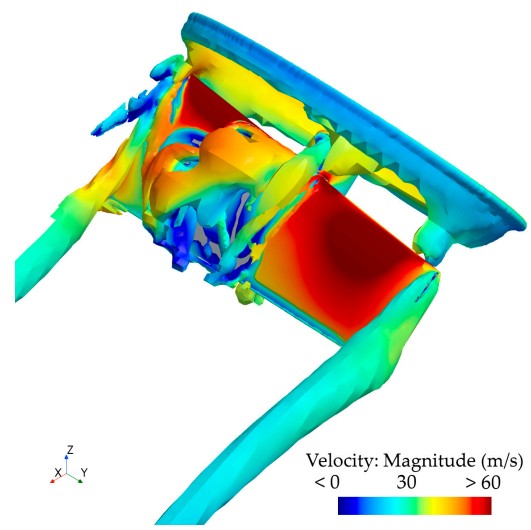

(**a**) Distribution of vortex system in tilt-wing during take-off transition section.

(**b**) Distribution of vortex system in tilt-wing during return transition section.

**Figure 17.** Comparison of the distribution of the vortices generated by the tilting wing during the take-off and the return transition sections at a flow velocity of 15 m/s and a tilt angle of 40°.

3.2.3. The Effect of Tilt Speed on the Take-Off transition Section at a Freestream Velocity of 30 m/s

The tilt velocity has a significant impact on the take-off transition section when the incoming flow velocity is 30 m/s. Figure 18a,b display the pressure coefficient distribution on the upper surface of the wing at tilt times of 10 s and 5 s, respectively, with a tilt angle of 49.5°. Figure 18c presents a contour of pressure coefficients on the upper surface of the wing obtained from a steady calculation with a tilt angle of 50°. When comparing Figure 18a–c, the negative pressure intensity on the upper surface of the tilt-wing is significantly lower at a tilt time of 10 s compared to 5 s. The pressure coefficient contour for a tilt time of 5 s is similar to the contour obtained from the steady calculation. Due to the stronger negative pressure on the wing surface at this tilt angle, the aerodynamic force at a tilt time of 5 s is greater than that at a tilt time of 10 s. Comparing the changes in the vortex system of the wing at this position can more intuitively reflect the reasons for this result.

The vortex distribution of the wing is plotted using the same method as in the previous section. Figure 19a–c depict the distribution of the wing's vortex system for tilt angles of 54°, 49.5°, and 45°, at the tilt time of 10 s, respectively. Figure 20a–c illustrate the distribution of the wing's vortex system for tilt angles of 54°, 49.5°, and 45°, at the tilt time of 5 s, respectively.

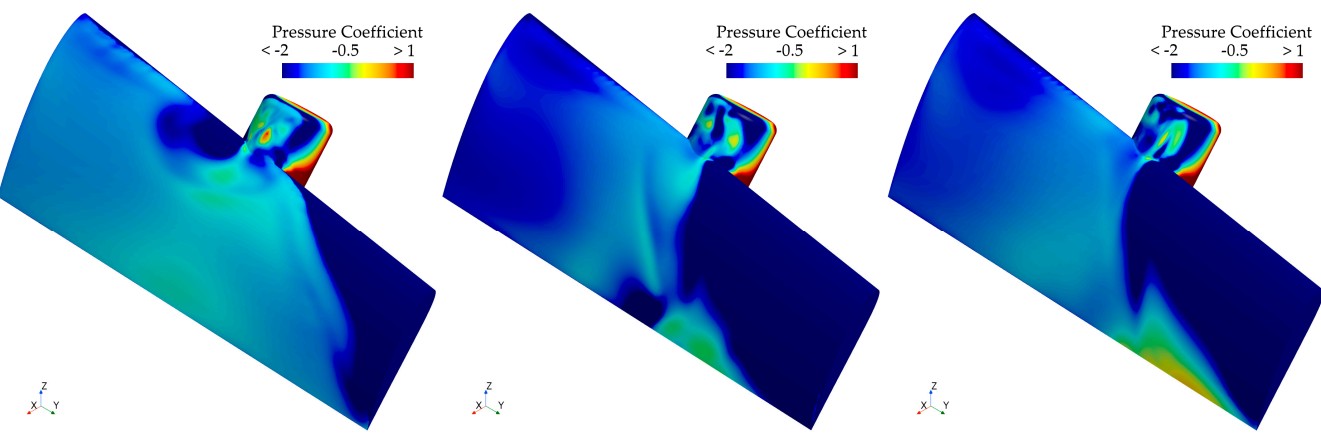

(**a**) Pressure coefficient on the upper surface of the tilt-wing at tilt angle 49.5° at a tilt time of 10 s.

(**b**) Pressure coefficient on the upper surface of the tilt-wing at tilt angle 49.5°at a tilt time of 5 s.

(**c**) Pressure coefficient on the upper surface of the tilt-wing at tilt angle 50° using steady calculation.

**Figure 18.** Pressure coefficient contour on the upper surface of the wing during take-off transition section at incoming flow velocity of 30 m/s and tilt angle 49.5° and 50°.

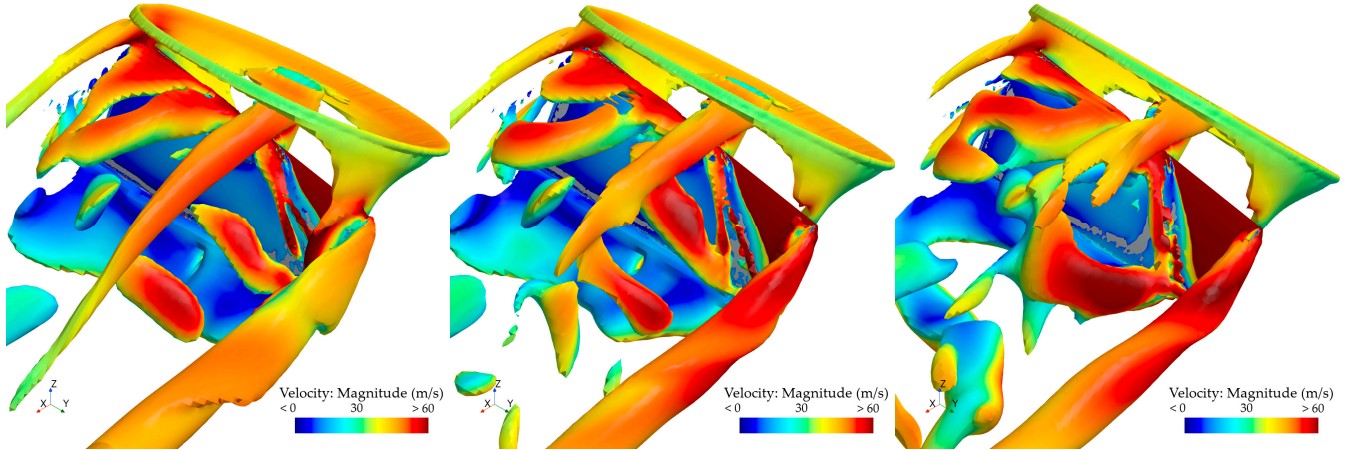

(**a**) Distribution of vortex system in tilt-wing at the tilt angle of 54° and a tilt time 10 s.

(**b**) Distribution of vortex system in tilt-wing at the tilt angle of 49.5° and a tilt time 10 s.

(**c**) Distribution of vortex system in tilt-wing at the tilt angle of 45° and a tilt time 10 s.

**Figure 19.** During the take-off transition section, with a completion time of 10 s and an incoming flow velocity of 30 m/s, the wing vortex distribution diagrams at tilt angles of 54°, 49.5°, and 45° are shown.

By comparing Figure 19 with Figure 20, it can be observed that the intensity of the vortex generated at the leading edge of the wing exhibited a significant reduction when the tilt duration was increased from 5 s to 10 s. As shown in Figure 19, on the side of the tilting wing near the fixed wing, the vortex system near the upper surface has been disrupted. The intensity of the vortex system generated at the leading edge of the wing is stronger when the tilt time is 5 s. Consequently, the vortex system remains intact at the same tilt angle. The flow on the upper surface of the wing is complex at this stage, and the vortex system develops more fully when the tilt speed is slower. When a high-energy vortex breaks up, the velocity of the upper surface of the wing decreases, resulting in a reduction in the negative pressure zone. This leads to a decrease in aerodynamic force.

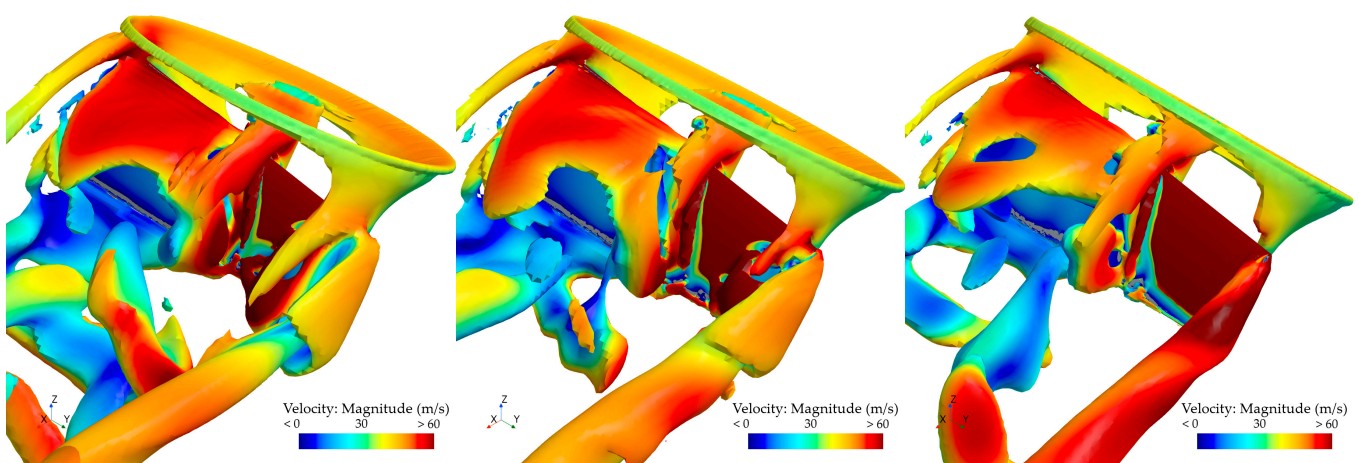

(**a**) Distribution of vortex system in tilt-wing at the tilt angle of 54° and a tilt time 5 s.

(**b**) Distribution of vortex system in tilt-wing at the tilt angle of 49.5° and a tilt time 5 s.

(**c**) Distribution of vortex system in tilt-wing at the tilt angle of 45° and a tilt time 5 s.

**Figure 20.** During the take-off transition section, with a completion time of 5 s and an incoming flow velocity of 30 m/s, the wing vortex distribution diagrams at tilt angles of 54°, 49.5°, and 45° are shown.

## 4. Conclusions

This paper establishes a grid suitable for studying the aerodynamic characteristics of the transition section of a tilt-wing aircraft. By using the momentum source method instead of the blade resolved CFD method, the computation time is significantly reduced. The influence of propeller slipstream on the aerodynamic characteristics of the wing in the transition section is studied using the RANS method at a fixed tilt angle. The aerodynamic characteristics of the transition section of the tilt-wing aircraft are simulated and analyzed using the RANS and URANS methods at fixed tilt angles and continuous tilting.

It was found that in the transition section of tilt-wing aircraft, the propeller slipstream effect can increase the lift, drag, and stall angle of the wing, but significantly reduce the lift-to-drag ratio. The above phenomenon will be weakened when the difference between the slipstream velocity and the incoming flow velocity decreases. The tilt-wing aircraft experiences strong nonlinearity and unsteadiness in the transition section. There are significant differences in the aerodynamic characteristics of the tilt-wing aircraft transition section between steady and unsteady calculations. Additionally, there are differences in the aerodynamic characteristics between the take-off and return transition section of the aircraft.

The above results indicate the following information:

(1) In the transition section, the slipstream velocity can be increased by adjusting the propeller speed, thereby allowing the aircraft to complete the mode transition faster.

(2) The take-off transition section requires a higher propeller speed compared to the return transition section to overcome the stall problem on the wing's upper surface.

(3) The tilt process and tilt speed of the tilt-wing aircraft affect the variation in aerodynamic forces in the transition section. If the aerodynamic data at a fixed tilt angle are calculated as the input for the flight control system, it may result in difficulties in maintaining altitude during transition section, which could affect flight safety.

**Author Contributions:** Conceptualization, Q.H. and G.H.; Methodology, Q.H. and G.H.; Software, Q.H., J.J. and F.Y.; Validation, Q.H., J.J. and Z.H.; Formal analysis, Q.H. and G.H.; Investigation, Q.H.; Data curation, Q.H. and G.H.; Writing—original draft, Q.H.; Writing—review and editing, G.H.; Visualization, Q.H.; Supervision, G.H.; Project administration, Q.H. and G.H.; Funding acquisition, Q.H. and G.H. All authors have read and agreed to the published version of the manuscript.

**Funding:** This work was partially supported by the National Natural Science Foundation of China (Grant No. 12362026) and Jiangxi Provincial Graduate Innovation Special Fund (Grant No. YC2022-s736).

**Data Availability Statement:** Data are contained within the article.

**Conflicts of Interest:** The authors declare no conflict of interest.

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
