# Peer review of "Numerical Simulation on Aerodynamic Characteristics of Transition Section of Tilt-Wing Aircraft"

_aerospace, doi:10.3390/aerospace11040283_

Round 1

Reviewer 1 Report

Comments and Suggestions for Authors

The article deals with the presentation of some aerodynamic characteristics of a generic "tilt-wing configuration" applying CFD.

The paper needs a thorough revision before it can be considered for publication. The following points should be carefully addressed:

1) Introduction. The introduction part is quite brief. There is numerous CFD work on rotor/propeller/wing configurations employing high-fidelity methods high-order time-accurate methods for such problems.

2) Geometry: A sketch along with the main dimensions of the chosen wing-propeller geometry must be shown. Further, the airfoil of the wing section should be mentioned. Design considerations with respect to propeller thrust and wing lift coefficient for the transition process should be highlighted. Further, the configuration geometry should be justified with respect to the chosen wing span section and propeller size (blade radius).

3) CFD: Standard RANS equations (Eq. 1-6) can be omitted as this can be taken from text books. But details of the (U)RANS method used herein are missing and must be provided: Which software has been used ? What is about discretization of convectice and viscous fluxes, spatial and temporal schemes, etc. There was some grid convergence study performed, what is about y+ values ? What is about time step convergence study ? A clear description of the numerical method used and the numerical set-up applied is mandatory.

4) Results: Typically, results should be presented using characteristics data like aerodynamic coefficients (lift, drag) instead of absolute values. Velocities should be referenced to the free stream velocity. 

Values/level for the pressure coefficient seems to be spurious, cf. Fig. 7b. Are there pressure coefficients in the order of +3 and of -10 ?

5) There should be a comment on novel findings taken from these investigations with respect to the state-of-the-art.

Comments on the Quality of English Language

A final proof reading and check on grammar and style is strongly recommended.

Author Response

Dear Reviewer,

Thank you sincerely for dedicating your time and effort to improving this manuscript. Your questions and comments were objective and professional, and they have greatly contributed to enhancing the content of this paper.

I have addressed the comments and advice you raised by making revisions to the manuscript. I have also provided detailed responses in the attached document for your reference.

Once again, I appreciate your critique and suggestions regarding the content of this paper.

Wishing you all the best.

Best regards,

Huang Qingjin

Reviewer 2 Report

Comments and Suggestions for Authors

Thank you very much for your submission. Please see attached word document with review comments. 

I recommend the following research to you, which is relevant to your manuscript:

This explores the use of more efficient CFD methods versus conventional rotor CFD methods, as described in the manuscript under review: “Efficient Computational Fluid Dynamics Approach for Coaxial Rotor Simulations in Hover” https://arc.aiaa.org/doi/full/10.2514/1.C036037

Author Response

(The authors gave the same response as above.)

Round 2

Reviewer 1 Report

Comments and Suggestions for Authors

Authors have addressed the concerns and remarks of the reviewer to a sufficient extent which is appreciated.

Comments on the Quality of English Language

A final proofreading is still recommended.

For example, sub-caption of Fig. 4: "simunation" -> simulation

Author Response

Dear Reviewer,

        Thank you very much for the time and effort you have dedicated to improving the quality of this manuscript. The previous revisions mainly focused on the content of the paper, which resulted in some oversights in terms of grammar and spelling. I sincerely apologize for this. In this revision, I have proofread the entire manuscript again, and I have also made the necessary corrections to the spelling error in Figure 4, as pointed out by you. I apologize once again for these simple errors that persisted even after the first round of revisions.

        Once again, I would like to express my gratitude for your efforts in enhancing the quality of this manuscript.

        Wishing you all the best.

Sincerely, 

Huang Qingjin

Reviewer 2 Report

Comments and Suggestions for Authors

Thank you for revising this manuscript.  I think it will be received well by the UAM community.  In the future, please do make efforts to compare your data with available experimental data to make your work even stronger.  There are several tilt-wing test programs being currently investigated throughout the community.  I think the conclusions that quasi-steady CFD could be used as a meaningful approximation in the transition region is very beneficial for the community to be aware of.  I think further work and investigation on the sensitivity to transition speed and RANS vs URANS simulation would be a good future avenue to pursue.  Thank you for your submission and hard work. 

Author Response

Dear Reviewer,

        Thank you very much for the time and effort you have invested in improving the quality of this manuscript. I find your suggestions to be highly constructive. The quasi-steady CFD simulation results indeed have significant reference value in the transitional section of tilting, which can greatly save computational resources and time. Furthermore, although the content of this paper stops here, my team is still conducting further research on the influence of tilt speed on the aerodynamic characteristics of tilt-wing aircraft.

        This revision mainly addresses the English grammar and spelling issues raised by another reviewer. I have once again proofread the entire manuscript for grammar and spelling.

        Once again, I greatly appreciate your highly professional and constructive perspectives and comments on this paper.

        Wishing you all the best.

Sincerely,

Huang Qingjin